# Differential B-Cell Receptor Signaling Requirement for Adhesion of Mantle Cell Lymphoma Cells to Stromal Cells

**DOI:** 10.3390/cancers12051143

**Published:** 2020-05-02

**Authors:** Laia Sadeghi, Gustav Arvidsson, Magali Merrien, Agata M. Wasik, André Görgens, C.I. Edvard Smith, Birgitta Sander, Anthony P. Wright

**Affiliations:** 1Department of Laboratory Medicine, Division of Biomedical and Cellular Medicine, Karolinska Institutet, 141 57 Stockholm, Sweden; laia.sadeghi@ki.se (L.S.); gustav.arvidsson@medsci.uu.se (G.A.); andre.gorgens@ki.se (A.G.); edvard.smith@ki.se (C.I.E.S.); 2Department of Laboratory Medicine, Division of Pathology, Karolinska Institutet, 141 52 Stockholm, Sweden; magali.merrien@ki.se (M.M.); agata.wasik@ki.se (A.M.W.); birgitta.sander@ki.se (B.S.); 3Institute for Transfusion Medicine, University Hospital Essen, University of Duisburg, 45 147 Essen, Germany

**Keywords:** B-cell receptor signaling, mantle cell lymphoma, ibrutinib, acalabrutinib, CXCR4, RNA sequencing, ICAM1, BTK, S1PR1, coculture

## Abstract

Interactions between lymphoma cells and stromal cells play a key role in promoting tumor survival and development of drug resistance. We identified differences in key signaling pathways between the JeKo-1 and REC-1 mantle cell lymphoma (MCL) cell lines, displaying different patterns of stromal cell adhesion and chemotaxis towards stroma-conditioned medium. The identified adhesion-regulated genes reciprocated important aspects of microenvironment-mediated gene modulation in MCL patients. Five-hundred and ninety genes were differently regulated between the cell lines upon adhesion to stromal cells, while 32 genes were similarly regulated in both cell lines. Regulation of B-cell Receptor (BCR) signature genes in adherent cells was specific for JeKo-1. Inhibition of BCR using siRNA or clinically approved inhibitors, Ibrutinib and Acalabrutinib, decreased adhesion of JeKo-1, but not REC-1 cells. Cell surface levels of chemokine receptor CXCR4 were higher in JeKo-1, facilitating migration and adhesion of JeKo-1 but not REC-1 cells. Surface levels of ICAM1 adhesion protein differ for REC-1 and JeKo-1. While ICAM1 played a positive role in adherence of both cell lines to stromal cells, S1PR1 had an inhibitory effect. Our results provide a model framework for further investigation of mechanistic differences in patient-response to new pathway-specific drugs.

## 1. Introduction

During the last decade, treatment approaches to hematological disorders have shifted from aggressive chemotherapy regimens towards specific targeted therapies that aim to reduce side effects and increase treatment efficiency. However, despite progress, there are problems to overcome for patients that are refractory to treatment or that relapse due to minimal residual disease or acquisition of drug resistance [1,2]. Mantle cell lymphoma (MCL) is a B-cell non-Hodgkin lymphoma (NHL) accounting for 7–9% of all lymphomas in Europe, and 6% in the United States [3,4]. MCL is characterized by the chromosomal translocation t(11;14)(q13;q32), which leads to aberrant cell cycle progression [5,6]. Most patients diagnosed with MCL are at advanced stage with extranodal involvement, including bone marrow, spleen, and the gastrointestinal tract [7,8].

MCL cells proliferate and survive in microenvironment niches, where they receive growth and survival signals from nonmalignant cells [9,10]. Interactions between MCL cells and their microenvironment are mediated by a complex network of cell adhesion molecules and cell surface signaling molecules as well as cytokines and their receptors [11,12,13]. Cells in the MCL microenvironment secrete cytokines, chemokines, and other soluble molecules that are important for tumor cell trafficking, homing, survival, and drug resistance. For example, stromal cells in secondary lymphatic tissues constitutively secrete chemokines, such as CXCL12, that facilitate homing and survival of tumor cells expressing its main receptor, CXCR4 [14,15]. MCL cells express high surface CXCR4 levels, and CXCR4 inhibition by the clinically approved antagonist, Plerixafor (also named AMD3100), reduces MCL cell migration and disrupts microenvironment interaction [16,17,18].

The B-cell receptor (BCR) is required for normal B-cell differentiation and maturation in different microenvironmental niches [19]. BCR signaling is mediated by multiple downstream signaling pathways, including phosphatidalyinositol-3-kinase (PI3K) and Bruton’s tyrosine kinase (BTK) [20]. BTK is essential for BCR signaling and for signal propagation to several downstream pathways [20,21] leading to activation of multiple kinases involved in cytokine release, cell migration, and cell adhesion [22,23,24].

The BCR is implicated in the pathogenesis of MCL where microenvironment-induced BCR activation enhances MCL survival and drug resistance [12,25]. Ibrutinib, previously called PCI-32765, is a clinically approved drug that covalently binds to BTK, inhibiting its kinase activity irreversibly [26,27]. Ibrutinib has been granted breakthrough therapy designation for the treatment of patients with MCL and chronic lymphocytic leukemia (CLL) [28]. In CLL, Ibrutinib causes redistribution of tumor cells from lymph nodes (LN) into the peripheral blood (PB) and inhibits their migration towards CXCL12 gradients [26,29,30]. Furthermore, Ibrutinib treatment of MCL reduced the secretion of the BCR-associated chemokines CCL3 and CCL4 and decreased the number of adherent cells in coculture [31]. For patients with relapsed or refractory MCL or CLL after conventional treatment, Ibrutinib is considered the drug of choice [32]. However, some patients do not respond to Ibrutinib treatment. It would therefore be important to identify predictive biomarkers to predetermine which patients will benefit from targeted therapies such as Ibrutinib.

Several in vitro systems have been established to model tumor microenvironment interactions that are important in B-cell lymphomas [16,33]. We recently developed a coculture system and gene expression analysis workflow that offers a way to systematically analyze microenvironment interactions between MCL cells and stromal cells in a controlled in vitro environment that can be used to study characteristics such as drug sensitivity [34]. We identified 1050 genes that are regulated differentially in stromal cell-adhered MCL cells compared to MCL cells in the suspension fraction of the coculture, and many of these genes are also differentially expressed in LN compared to PB in MCL and CLL patients [23,34,35]. Thus, the model system reflects important aspects of microenvironment interactions observed in patients and facilitates specific modeling of highly relevant processes such as microenvironment-mediated drug resistance. Using this model system, we sought to identify signaling pathways required for adhesion of MCL cells to stromal cells in MCL cells of different origin. To this end, we comprehensively compared global transcript level changes in two different MCL cell lines upon adhesion to stromal cells and here report differential gene expression profiles that differ depending on the origin of the tumor cells.

## 2. Results

### 2.1. Different Adhesion of JeKo-1 and REC-1 MCL Cells to Stroma

JeKo-1 cells adhere strongly to stromal cells (MS-5) in coculture [34]. To compare REC-1 to JeKo-1, each of the cell lines were cocultured with MS-5 cells and the ratio of bound MCL cells to stromal cells was analyzed at different time points using flow cytometry of fluorescently labeled cells (Appendix A and Appendix A). Like JeKo-1, REC-1 cells bound stromal cells stably, but the ratio of adhered REC-1 cells to stromal cells quickly reached a plateau following an initial increase, while for JeKo-1, stromal adhesion continued to increase at later times (Figure 1A). There was no apparent difference between JeKo-1 and REC-1 cells with regard to strength of binding to stromal cells after 4 h, as judged by resistance to vigorous agitation (Appendix A).

Transwell assays for quantification of cellular migration indicated that JeKo-1 cells migrated more efficiently than REC-1 cells, while both JeKo-1 and REC-1 migrated more efficiently towards conditioned medium than to medium alone (Figure 1B). In both settings, we could exclude that the lower migration and adhesion capacity of REC-1 cells was due to reduced viability of REC-1 cells (Appendix A). We concluded that JeKo-1 and REC-1 cells likely use different mechanisms for microenvironment communication and hypothesized that those mechanisms could be revealed by global gene expression profiling.

### 2.2. Adhesion to Stroma Affects Global Gene Expression Differently in JeKo-1 and REC-1 Cells

mRNA was extracted from JeKo-1 and REC-1 cells after 24 h coculture with MS-5 cells. To omit time-consuming cell separation procedures, shown to artefactually induce changes in mRNA levels, RNA from lymphoma cells adhered to stromal cells was extracted and sequenced to produce mixed-species cDNA libraries and sequence reads that were subsequently deconvoluted in silico, as has been described previously [34]. Global transcript level changes were subsequently calculated between nonadherent suspension (Susp) and adherent (Adh) MCL cells within the cocultures. Monocultured cells (Sep) from both cell lines were included as controls. Principle component analysis indicated that while JeKo-1 and REC-1 are two cell lines representing the same type of hematological tumor, their gene expression profiles are distinct, as shown by separation along the first principal component (Figure 2A). Differences between Sep, Susp, and Adh are shown by the second principal component for both cell lines and the broader spread of the JeKo-1 samples indicates a stronger differential regulation of genes between different coculture conditions.

In total, 549 and 291 genes with significantly altered transcript levels between Adh and Susp cells were identified for JeKo-1 and REC-1, respectively (false discovery rate (FDR) q-value ≤ 0.05, fold change ≥ 1.5, Figure 2B and Appendix A). Surprisingly, only 34 genes were common to both sets of differentially regulated genes. Appendix A shows that this set of genes is significantly enriched in oxidative phosphorylation KEGG pathway components (e.g., *MT-CO1*, *MT-CO2*, *MT-CO3*, *MT-ND2*, and *MT-ND3*) and their lower expression in adherent cells (Figure 2C) is consistent with the low metabolic rate associated with cellular quiescence seen in adhered MCL and other NHL cells [36]. Also in the intersect were *TRAF1*, which increases in MCL cells upon adhesion to stromal cells to protect the MCL cells from undergoing apoptosis [33] (Figure 2C), and the transcription factor gene *HOXA13*, which is epigenetically silenced in several MCL cell lines [37].

A total of 590 genes showed different regulation upon adhesion to stroma between JeKo-1 and REC-1 cells (FDR *q*-value ≤ 0.05, Figure 2D). As indicated by the colors in the volcano plot in Figure 2E, the majority of corresponding transcripts had a greater fold change in JeKo-1 (405/590, pink) than in REC-1 (185/590, yellow). Notably, several of the genes among those with large variation in differential expression between the two cell lines included the adhesion molecule ICAM-1 and molecules important for chemotaxis, such as CXCR4 and CSF1, which are highly expressed in MCL cells [38], as well as CCL4, which is known to increase upon BCR engagement [39]. Other genes with a significant differential change in expression included *EGR1*, which also functions downstream of the BCR, as well as components of the canonical NF-κB pathway, such as *NFKB1* and *NFKBIE*. Gene set enrichment analysis (GSEA) using pathways in the KEGG database identified significant enrichment of genes involved in Toll-like receptor, T-cell, and B-cell receptor signaling pathways, among others in genes that were more highly induced in adherent JeKo-1 cells than adherent REC-1 cells (Figure 2F and Appendix A). No significantly enriched pathways corresponding to genes that were more highly regulated in REC-1 cells were found. Thus, adhesion to stromal cells appears to lead to a greater engagement of signaling events in JeKo-1 than in REC-1 cells.

In a previous analysis of the JeKo-1 RNA-seq data, we showed that adhesion-regulated genes encode proteins containing a significantly higher proportion of intrinsically disordered regions (IDRs) [40]. IDRs have been associated with a greater capacity for functional adaptation [41] as well as being enriched in genes with low expression levels, which are known to be enriched in protein interaction domains with the capacity to interact with multiple partner proteins [42]. It was therefore of interest to determine whether there was a difference in IDR content between proteins encoded by genes with a similar or different regulation during adhesion in JeKo-1 and REC-1 cells as well as whether there was a correlation of gene expression level and IDR content of encoded proteins for either of these gene sets. Appendix A shows that the gene set that is similarly regulated in JeKo-1 and REC-1 cells during adhesion has a significantly lower IDR content than differently regulated genes (*p* = 0.0015). Furthermore, there is a strong negative correlation between expression level in JeKo-1 cells and IDR content of encoded proteins for the set of genes that are similarly regulated in both cell lines (rho = 0.8, *p* = 4.01 × 10^−6^, Appendix A), whereas there is only a very low, albeit significant, correlation to the expression levels of the same genes in REC-1 cells (Appendix A). The number of genes analyzed in this set (*n* = 23) is rather low, and thus the functional significance of the difference between JeKo-1 and REC-1 cells is uncertain. It may also be important to note that the nine very highly expressed genes encoding proteins with very low IDR content consist of a group of mitochondrial enzymes involved in oxidative phosphorylation. For the larger number of genes that are differently regulated between the cell lines (*n* = 288), there is no significant correlation of expression level with regard to IDR content for either cell line (Appendix A).

### 2.3. Adhesion-Regulated Gene Expression in Cell Lines Mimics Microenvironment-Regulated Gene Expression in MCL Patients

To evaluate how well adhesion-regulated gene expression in JeKo-1 and REC-1 cells corresponds to microenvironment-mediated gene regulation in patients, we used published patient data in order to compare transcript-level differences between suspension and adherent JeKo-1 or REC-1 cells in vitro with gene expression differences between LN- and PB-derived cells from previously untreated MCL patients (*n* = 8) [23,34,35]. Our analysis showed that 684 adhesion-related genes in cocultured JeKo-1 or REC-1 cells overlapped with the set of genes in MCL patients that were differentially expressed between LN and PB. Clustering of this set of 684 genes showed that adhesion-regulated genes in JeKo-1 and REC-1 cells correspond to a set of microenvironment-regulated MCL tumor cells from patients. Thus, with respect to adhesion-regulated gene expression, there is a high degree of similarity between the cell lines and a major subset of patient samples in relation to the variance within the patient group (Figure 3A,B). We found no obvious explanation for the apparent clustering of the patient samples into two separate groups, but this is a finding that should be further explored. We conclude that JeKo-1 and REC-1 cells are a relevant model system for studying aspects of microenvironment-regulated gene expression in MCL.

### 2.4. Differences in Gene Expression Changes for BCR Signature Genes in JeKo-1 and REC-1 Cells Upon Adhesion to Stromal Cells

Enrichment of gene sets with differential expression in adherent JeKo-1 and REC-1 cells, identified by GSEA, as well as identification of several genes in this set that function downstream of the BCR (e.g., *CD83*, *EGR3*, *DUSP2*, *NFKB1*, and *CCL4,*
Figure 2E,F and Appendix A) led us to further investigate BCR signaling, which plays an essential role in the pathogenesis of MCL [43]. The median expression level for a list of previously published BCR signature genes was significantly increased in JeKo-1 upon adhesion to stromal cells, while there was no difference for REC-1 cells (Figure 4A,B) [23]. Consistently, transcript levels for NF-κB signature genes increased in JeKo-1 upon adhesion to stromal cells [29,33] (Appendix A), but not in REC-1 cells. Thus, there is a difference between adherent JeKo-1 and REC-1 cells with regard to activation of the BCR and related pathways.

Upon BCR stimulation by anti-human IgM, rapid phosphorylation of downstream targets, BTK, and PLCγ2 was detected in both JeKo-1 and REC-1 cells, indicating that both cell lines express a functional BCR (Figure 4C and Appendix A). However, only JeKo-1 showed increased levels of the BCR regulated CCL3 and CCL4 cytokines upon coculture, and at similar levels to those induced by anti-human IgM treatment of monocultured cells (Figure 4D). Taken together, these results indicate that REC-1 cells have a functional BCR with a higher potential to activate at least some downstream targets, compared to JeKo-1 cells, but that BCR is not engaged, as it is in JeKo-1 cells, upon adhesion of REC-1 cells to stromal cells.

### 2.5. BCR Signaling is Important for Adhesion of JeKo-1 but not REC-1 Cells to Stroma

Consistent with the BCR-related results above, treating cocultured cells with the BTK inhibitor Ibrutinib, or transiently knocking down BTK using siRNA (Figure 4E, Appendix A and Appendix A), reduced the number of adherent cells for JeKo-1 but not REC-1, which are apparently refractory to treatment with BCR inhibitors in this respect (Figure 4E). To determine whether BTK is required for the adhesion process per se and/or for maintaining stromal adhesion, JeKo-1 cells were treated with Ibrutinib and the second generation BTK inhibitor Acalabrutinib [44], either for 1 h prior to wash out and then coculture or after 24 h of coculture with stromal cells. Both treatment protocols reduced the number of adhered cells, indicating that BTK is required both for the establishment and maintenance of adherent interactions with stromal cells (Appendix A). This might imply the existence of more than one BCR-regulated mechanism. MS-5 cells do not express BTK and treatment of the stromal cells with Ibrutinib or Acalabrutinib as well as BTK knock-down prior to coculture experiments had no significant effect on MCL cell adhesion (Figure 4C, Appendix A). No BTK enzymatic activity was detectable after 1 h incubation with Ibrutinib or Acalabrutinib, and no restored activity was detected during the 4 h that followed removal of the compounds (Appendix A and Appendix A), thus extensive inhibition was obtained at the inhibitor levels used and there was no significant resynthesis of BTK during the time courses of the experiments in (Appendix A). Taken together, the BCR appears to have functional roles in both establishment and maintenance of JeKo-1, but not REC-1, cell adhesion to stromal cells even though REC-1 has a functional BCR.

### 2.6. JeKo-1 and REC-1 Cells Express Different Cell Surface Levels of CXCR4

The G-protein coupled cell surface receptor, encoded by *CXCR4*, was among the most differentially expressed genes in adherent cells between JeKo-1 and REC-1 (Figure 2E). Nonadherent JeKo-1 cells exhibited higher cell surface levels of CXCR4 compared to their REC-1 counterparts (Figure 5A). Cell surface levels of CXCR4 were reduced significantly in stromal cell-adhered JeKo-1 cells compared to suspension cells. Higher CXCR4 surface levels in suspension coculture JeKo-1 cells could account for their more efficient migration compared to REC-1 cells (Figure 1B). To test this, we used Plerixafor to reduce cell surface levels of CXCR4 (Figure 5A). Plerixafor treatment significantly inhibited migration towards the CXCR4 ligand, CXCL12, (Figure 5B), and adhesion to stromal cells (Figure 5C) for JeKo-1 cells, but not REC-1 cells. Figure 5D shows that CXCR4 surface levels are essentially reduced to background levels at the concentration of Plerixafor used (25 µg/mL). Since inhibition of either BCR or CXCR4 signaling impairs adhesion of JeKo-1 cells (Figure 4E–Figure 5C) we evaluated the existence of a connection between CXCR4 and BTK signaling. There was little or no additional effect of Plerixafor in cells that were fully inhibited by the irreversible BTK inhibitor, Ibrutinib, suggesting that the BCR and CXCR4 may affect steps in the same process that leads to JeKo-1 cell adhesion (Figure 5E).

### 2.7. ICAM1 and S1PR1 are Important for the Adhesion of MCL Cells to Stroma

*ICAM1,* encoding an adhesion molecule important for immune responses, where it promotes leukocyte adhesion to the endothelium of the vascular wall [45], was regulated differently in JeKo-1 and REC-1 cells during adhesion to stromal cells (Figure 2E). The ICAM1 cell surface level was higher in nonadherent REC-1 cells compared to JeKo-1, but its level is oppositely regulated in the two cell lines and converges to a similar level upon adhesion to stromal cells (Figure 6A). siRNA knock down of ICAM1 significantly impaired adhesion of both JeKo-1 and REC-1 cells, indicating an important role in adhesion to stromal cells (Figure 6A,B and Appendix A).

To more broadly evaluate genes that could contribute to adhesion, 32 genes, including *CD48, CD79A, IL10RA, PLEK*, and *S1PR1,* were identified that are common to both the 560 genes whose differential expression differed most between the cell lines and the publicly available OKCAM database of adhesion-related molecules (Figure 6C) [46]. All of these 32 genes have been coupled to adhesion and/or migration of hematopoietic cells [46]. The common genes also contain cadherin and protocadherin genes, with known roles in some cell interactions, but these are downregulated in adherent JeKo-1 cells and thus were not considered further. The common gene list contained the sphingosine-1-phosphate receptor (*S1PR1*). In mice, increased S1PR1 levels promote lymphocyte mobilization from secondary lymphoid organs into blood [47]. S1PR1 expression is reduced in JeKo-1 cells upon adhesion and is expressed at constitutively higher levels in REC-1 cells (Figure 6C). Higher S1PR1 transcript levels in REC-1 as compared to JeKo-1 have been reported previously [48]. It is possible that the low adhesion capacity of REC-1 cells is due, in part, to higher S1PR1 expression levels that might enhance detachment from stromal cells. Consistently, siRNA knock down of S1PR1 expression levels in JeKo-1 and REC-1 cells (Appendix A) significantly increased both JeKo-1 and REC-1 cell adhesion to stromal cells (Figure 6D).

## 3. Discussion

Using an in vitro coculture model system that mimics important aspects of tumor microenvironment interactions observed in MCL patients, we show that two MCL cell lines, JeKo-1 and REC-1, use distinct signaling pathways and molecules to communicate with the microenvironment. A total of 590 genes were identified to be differently changed between stroma-adherent JeKo-1 and REC-1 cells relative to suspension cells in coculture. BCR-regulated genes and CXCR4 were specifically important for migration and adhesion of JeKo-1 cells, while ICAM1 and S1PR1 affected adhesion of both cell lines to stromal cells. Importantly, the adhesion of JeKo-1 cells, but not REC-1 cells, to stromal cells required BTK-dependent signaling downstream of the BCR and was sensitive to clinically used BTK inhibitors such as Ibrutinib and Acalaibrutinib. If this difference is also seen in MCL patients, it would provide a potential explanation for why some patients are refractory to Ibrutinib treatment, while others respond. Potentially, therefore, a simple cell adhesion test and its sensitivity to BTK inhibitors could be used to classify patients prior to treatment.

During B-cell development, CXCR4 regulates B-cell mobility and migration between lymphoid organs and the PB [49]. CXCR4 expression is upregulated in CLL and MCL, affecting migration and homing of malignant B-cells to lymphoma niches [16,50]. Consistent with previous studies, we showed that JeKo-1 cells express functional cell surface CXCR4 that regulates migration towards, and adhesion to, stromal cells [16]. CXCR4 levels were low on REC-1 cells, and CXCR4 was not required for stromal adhesion. These differences may reflect the different origins of the cell lines. JeKo-1 cells originate from PB and REC-1 cells from LN and it is known that MCL/CLL cells in PB express higher levels of CXCR4 compared to those in LN [29,31].

MCL stromal adhesion led to a reduction of CXCR4 surface levels, consistent with the lower CXCR4 surface levels reported for LN-resident CLL cells compared to those in PB [29]. Furthermore, CXCR4 inhibition prevents migration of MCL cells beneath the stromal cells and abrogates microenvironment protection [16], making the tumor cells more accessible to chemotherapy. The effect of Plerixafor on JeKo-1 cells presented here reflects in vivo observations seen in Acute Myeloid Leukemia (AML) and CLL [51,52], but the refractory nature of REC-1 cells suggests that this therapy approach may not be applicable to all MCL cases.

In addition, CXCR4 and S1PR1 antagonize each other in order to modulate the balance between B-cell homing and retention within microenvironmental niches versus mobilization into PB [53]. High surface S1PR1 levels facilitate mobilization of B-cells from these niches into the circulation. MCL cells express high surface levels of S1PR1 [48], and low S1PR1 expression contributes to retention within microenvironmental niches [54], likely promoting tumor cell survival. Consistently, our results showed that S1PR1 is expressed on both JeKo-1 and REC-1 cells, albeit at higher levels on REC-1 cells, as shown previously [48], and its attenuation enhanced the capacity for JeKo-1 and REC-1 cells to adhere to stromal cells. Different S1PR1 levels on JeKo-1 and REC-1 cells might be associated with their tissue of origin, since B-cells express different S1PR1 levels depending on their differentiation state and localization [48]. Moreover, overexpression of S1PR1 reduces surface levels of CXCR4, leading to an impaired migratory response [55] and Plerixafor injection in mice increases S1PR1 surface expression on progenitor cells in the bone marrow, which enhances their mobilization and migration towards the PB [56].

Cellular/stromal adhesion of immune cells is a complex process mediated by cadherins, integrins, and other adhesion related molecules [57]. Here, we focused on genes regulated during adhesion to stromal cells and identified ICAM1 to be selectively upregulated in adherent JeKo-1 cells. ICAM1 is normally present at the cell surface of endothelial cells and some lymphocytes and monocytes, and a subgroup of patients with NHL including MCL and diffuse large B cell lymphoma (DLBCL) were previously described to express ICAM1 [58,59]. ICAM1 is the ligand for leukocyte function-associated antigen1 (LFA1, also known as ITGB2) and plays an important role in cell-to-cell adhesion. ICAM1 levels additionally correlate with the metastatic stage of lymphoma dissemination and clinical stage of CLL [58,59,60].

Both mRNA and cell-surface levels of ICAM1 were elevated in REC-1 compared to JeKo-1 cells; however, knock-down results confirmed the importance of ICAM1 for cellular adhesion of both cell lines. This indicates an important role of ICAM1 in microenvironment interactions that could account for reports that high-grade lymphomas express higher levels of ICAM1 compared to low-grade tumors [61]. Moreover, MCL and CLL cells have higher expression levels of ICAM1 in LN compared with PB [34]. VLA-4 is an adhesion molecule that is often highly expressed on MCL cells, but it was not among the differentially expressed genes in this study [16].

Activation of BCR triggers signaling events leading to B-cell proliferation, differentiation, and antibody production. In B-cell malignancies, such as CLL, MCL, and DLBCL, BCR signaling is important for pathogenesis and disease progression [20,22], including remodeling of microenvironments to favor survival of tumor cells [38]. In the latter study, CSF-1 and IL-10 produced by MCL cells were associated with polarization of macrophages, a process that was abrogated by Ibrutinib treatment. Since both CSF-1 and IL-10 are highly induced upon adherence of JeKo-1 MCL cells to stromal cells, one possible explanation for the effect of Ibrutinib on macrophage polarization could be the Ibrutinib-mediated inhibition of MCL cell adherence to stromal cells observed here. Ibrutinib reduces BCR signaling by inhibiting BTK and is a promising alternative for treating MCL, CLL and other B-cell malignancies [30,62]. However, some MCL and CLL patients are refractory to Ibrutinib treatment [12,63]. Here, we showed that JeKo-1, but not REC-1 MCL cells, require BCR signaling for their migration and adhesion to stromal cells, suggesting that REC-1 cells have adapted non-BCR-mediated survival strategies. REC-1 cells could therefore represent a class of MCL cells that in a clinical context would be refractory to Ibrutinib and other BTK inhibitors that are now being used clinically even though, like REC-1, such cells might have a functional BCR system.

Comparative analysis of our coculture data with data from eight MCL patient samples isolated from LN and blood [23,35] revealed that JeKo-1 and REC-1 cells clustered with a group of four MCL patient samples. Although there is no obvious explanation for the apparent clustering of the MCL patient samples into two separate groups, it is interesting to note that clustering of differentially expressed microenvironment/adhesion related genes led to separation of MCL patient samples into two groups. The identification of the responsible genes in the previously published data set and their characterization would be of interest, but is beyond the scope of this paper. The clustering indicated that JeKo-1 and REC-1 cells do not vary greatly from primary MCL cells with regard to gene expression changes associated with stromal cell adhesion. Comparisons of global gene expression data from the present study with clinical studies support the significance of this coculture model system and its use in mechanistic studies and preclinical studies of novel candidate drugs targeting MCL cells.

In vitro coculture systems, such as the one presented here, can enhance our understanding of how the tumor microenvironment influences disease progression and development of drug resistance. The results showed that the JeKo-1 and REC-1 cell and their interactions with stromal cells are relevant models for the aspects of MCL microenvironment signaling detected in clinical samples that are studied here. The results provide a foundation for future ex vivo and in vivo studies, which lie beyond the scope of this mechanistic study, for instance by single cell RNA-sequencing or by spatial transcriptomics of primary material from, e.g., LN in order to investigate gene expression differences within the cellular composition of microenvironments in patients. Future studies could also include reconstitution of more complex coculture systems containing more than two cell types.

## 4. Materials and Methods

### 4.1. Cell Lines, Reagents, and Antibodies

Cells were cultivated at 37 °C and 5% CO^2^ in media supplemented with 100 U/mL penicillin and 100 μg/mL streptomycin. The mouse stromal cell line MS-5 was purchased from DSMZ (German Collection of Microorganisms and Cell Culture GmbH). HS-5 human stromal cell line was obtained from ATCC (American Type Culture Collection). The MCL cell line JeKo-1 was purchased from DSMZ and REC-1 was kindly provided by Dr. Christian Bastard (Ronan, France). HS-5, MS-5, and MCL cell lines were maintained in DMEM (Gibco, NY, USA) or αMEM-glutamax (Gibco) or RPMI-glutamax (Gibco) respectively supplemented with 10% heat-inactivated fetal bovine serum (HI FBS, Gibco). Cocultures of JeKo-1/REC-1 with MS-5 and HS-5 at a 10:1 ratio was maintained under the same conditions as for MS-5 or HS-5 cells alone. Western blot (WB) reagents including the SDS-PAGE (4–12% Tris-Glycine) gels and Nitrocellulose membranes of iBlot Dry-blotting system were purchased from Life technologies (NY, USA). Anti-actin (1:100,000, Sigma, St. Louise, MO, USA), anti-BTK (1:100, HPA001198, Sigma), anti-phospho-BTK (Y551) (1:100, 558034 BD Pharmingen, USA), anti-phospho-BTK (Y223) (1:100, ab68217 Abcam, San Francisco, USA), anti-PLCγ (1:100, Sc-407 Santa Cruz Biotechnology, Santa Cruz, CA, USA), and anti-phospho-PLCγ (Y579) (1:100, ab75659 Abcam) antibodies were used for western blotting. PCI-32765 (Ibrutinib) and ACP-196 (Acalabrutinib) were purchased from Selleckchem (USA). AMD3100 (Plerixafor) was purchased from Sigma (A5602).

### 4.2. Cell-Cell Binding Assay and Flow Cytometry

CellTracer Carboxyfluorescein succinimidyl ester (CFSE, C34554 Thermo Fisher Scientific, TMO, NY, USA) and Far red (C34564 Thermo Fisher Scientific) were used to label the JeKo-1/REC-1 and MS-5/HS-5 cells, respectively, according to manufacturer’s instructions. To calculate the optimal concentration of Far red, a titration experiment was performed. CFSE (0.5 μM) labeled JeKo-1/REC-1 suspension cells were added to established Far red labeled MS-5 monolayers. After indicated time points, suspension JeKo-1/REC-1 cells were removed and monolayer MS-5 cells together with adherent JeKo-1/REC-1 cells were washed twice with PBS and trypsinized for 1 min. Then cells were stained with 4′,6′-diamidino-2-phenylindole (DAPI) and subjected to flow cytometry analysis. For cell surface receptor analysis, cells were incubated with either mouse anti-human PE-conjugated CXCR4 (FAB173P, R&D Systems, Minneapolis, MN, USA), or with mouse anti-human BV510-conjugated ICAM1 (740170, BD Bioscience, San Jose, CA, USA) antibody for 30 min at room temperature, followed by washing twice with cold wash buffer containing PBS and 0.5% FBS. Flow cytometry data were obtained with a MACSQuant Analyzer 10 (Miltenyi Biotec, Germany), and analyzed with FlowJo software (FlowJo version 10, USA).

### 4.3. siRNA Transfection and Real-Time Quantitative PCR

JeKo-1 or REC-1 cells were transfected with 500 nM specific ICAM1, BTK, S1PR1 (cat# 4390824, s4447, Invitrogen, CA, USA), or control siRNA (Life technologies, NY, USA) using the Neon transfection system (Thermo Fisher Scientific, TMO, NY, USA) (1650 V, 13 ms, 2 Pulses). Total RNA from JeKo-1 or REC-1 cells was isolated using TRIzol (Thermo Fisher scientific) followed by DNaseI treatment (Fermentas, MA, USA). A 1 μg sample of RNA was reverse-transcribed into cDNA using High-Capacity Reverse Transcription kit (Thermo Fisher Scientific). Real-time quantitative PCR was performed with 2 μL of cDNA using 2× SYBR Green Master Mix (Applied Biosystems, Carlsbad, CA, USA).

### 4.4. RNA Extraction, Library Preparation, and Sequencing

Mono and cocultures were set up as previously described [34]. Total RNA was extracted following 24 h of coculture using RNeasy kit (Qiagen, California, USA) with QIAshredders (Qiagen) according to manufacturer instructions. Following quality assessment by TapeStation, libraries were prepared using TruSeq RNA Library Prep Kit v2.0 (Illumina, San Diego, CA, USA) according to the manufacturers protocol with mRNA enrichment by poly-T oligo-attached magnetic beads. The libraries were sequenced using an Illumina HighSeq 2500 instrument (San Diego, CA, USA) generating on average 13 million 2 × 101 bp paired-end reads per sample.

### 4.5. Species-Based Read Separation and Mapping to Reference Genomes

Reference genomes GRCh38.87 and GRCm38.87 with corresponding annotation (.gtf) files were downloaded from ENSEMBL (Wellcome Genome Campus, Hinxton, UK). Indices were built for the species-based short read separation software Xenome (v1.0.1) (Monash University, Clayton, Australia) [64] and the short-read aligner STAR (v2.5.1b) (available online: http://code.google.com/p/rna-star/.) [65]. Raw paired-end short reads were divided based on species origin using the Xenome software with default settings. Separated, paired-end short reads of human and murine origin were aligned to reference genomes GRCh38 and GRCm38, respectively, using the splice aware short read aligner STAR (v2.5.1b) [65] with default options and –sjdbGTFfile pointing to the ENSEMBL.gtf annotation file corresponding to the reference genome and –outFilterMultimapNmax set to 1. Fragments per feature (concatenated exons for one gene) were determined using featurecounts from the package Subread (v1.5.2) (CA, USA).

### 4.6. Differential Gene Expression Analysis and GSEA

Differential gene expression between fractions of JeKo-1 and REC-1 cells was determined using the Bioconductor package edgeR (v3.22.5) (Vincent´s Institute of medical research, Victoria, Australia) following the glmQLF workflow [66]. Genes were ranked on log2 transformed fold changes multiplied by the negative log10 transformed adjusted *p*-value and subsequently subject to gene set enrichment analysis using the Bioconductor package fGSEA (v1.6.0) (ITMO, St Petersburg, Russia) for KEGG pathway gene sets (c2.cp.kegg.v6.2.symbols.gmt) downloaded from the MSigDB (http://www.broadinstitute.org/gsea/msigdb/index.jsp) (Broad Institute, MA, USA) [67].

### 4.7. Data Availability

The datasets generated and analyzed are available for download via the gene expression omnibus (GEO) [68] repository, by accession numbers GSE99501 (JeKo-1 data) and GSE122739 (REC-1 data).

### 4.8. Microarray Data Analysis of Publicly Available Data Sets

Affymetrix microarray gene expression data for cells of lymph node (LN) and peripheral blood (PB) origin from MCL patients was downloaded from the gene expression omnibus using accession numbers GSE70910 and processed as previously described [34]. Intersects between sets of differentially expressed genes were evaluated by Fisher exact tests and subsequently subjected to GO term enrichment analysis using the Bioconductor R package clusterProfiler (v3.10.1, gene sets for biological processes with >10 and <1000 genes were included in the analysis) (Lucent technologies, NJ, USA). Probe sets with reliable expression levels (log2(intensity) >4) for all samples in at least one condition were included in the analysis. Gene expression changes between LN and PB were calculated for each patient alongside fold changes determined by edgeR for adherent and suspension cells in coculture for JeKo-1 and REC-1 and were used for hierarchical clustering (R package: pvclust v2.0-0, method.dist = “euclidean”, method.hclust = “average”, nboot = 1000) or tSNE analysis (R package: Rtsne v0.15, perplexity = 8, theta = 0.5).

### 4.9. Cytokine Levels in Conditioned Media

MS-5 stromal cells were seeded 24 h in advance onto tissue culture treated 12 well plates at 5 × 10^4^ cells per well. 25 × 10^4^ JeKo-1/REC-1 cells were subsequently added to the stromal cells or to separate wells supplemented with anti-IgM at 1 µg/mL or 10 µg/mL (unlabeled goat F(ab’)2 Anti-Human IgM, Southern Biotech, AL, USA). Monocultured MS-5 and JeKo-1/REC-1 cells were seeded as controls. At 24 and 48 h, cells were spun down and conditioned media was stored at −20 °C until further analysis. The samples were diluted 1:5 using diluent buffer provided by kit and analyzed by sandwich ELISA for human CCL3 and human CCL4 according to the manufacturer’s instructions (R&D Systems, DMB00 and DMA00).

### 4.10. Cell Migration Assay

MS-5 stromal cells were seeded 24 h in advance as described above, and conditioned medium from the culture (CM) was collected at the time of experiment. JeKo-1 and REC-1 cells (cell density of 0.8 × 10^6^ cells/mL) were stained with calcein-AM (1 µM; Thermo Fisher Scientific, NY, USA) for 30 min at 37 °C, 5% CO2. Cells were washed twice and resuspended to 2 × 10^6^ cells/mL density in either 10% FBS in αMEM-glutamax in the case of chemotaxis towards CM or in 10% FBS in RPMI-glutamax in the case of chemotaxis towards CXCL12. Chemotaxis assay was performed using Boyden chamber system (Corning, NY, USA), during which cells were placed on top of the Fluoroblok insert (8 µm pore size; Corning, NY, USA) with the well below containing either medium alone, CM or medium with CXCL12 (200 ng/mL; R&D Systems, Minneapolis, MN, USA). Cumulative numbers of migrated cells were acquired using a Nikon Eclipse Ti confocal microscopy system (Nikon Instruments, Melville, NY, USA) with a 10× objective. Migration was quantified by counting fluorescent cells in bottom wells with NIS-Elements AR software (Nikon Instruments, Melville, NY, USA) either in a time lapse manner or at a single time point (4 h). When indicated, cells were treated with AMD3100 (25 µg/mL) (Sigma, St. Louise, MO, USA) 20 min prior to placing the cells in the upper chamber.

### 4.11. Statistical Analysis

All the data are reported as the sample mean ± the standard error of the mean (SEM). Either pairwise comparisons between means of different treatments of same cell type or unpaired comparison between mean of groups were performed using a Student *t*-test (two-tailed, paired/unpaired) for each couple of normally distributed populations. Throughout the text, significant statistical differences are indicated for *p* values.

## 5. Conclusions

Here, we propose that functional in vitro assays based on MCL stromal adhesion can identify and model relevant pathways and might be used to identify patients likely to respond to novel treatments. Importantly, such model systems will improve our understanding of potential pathological strategies employed in different cases of MCL and facilitate the identification of potential biomarkers that could be used to improve decision making in relation to which patients should be offered which treatments.

## Figures and Tables

**Figure 1 cancers-12-01143-f001:**
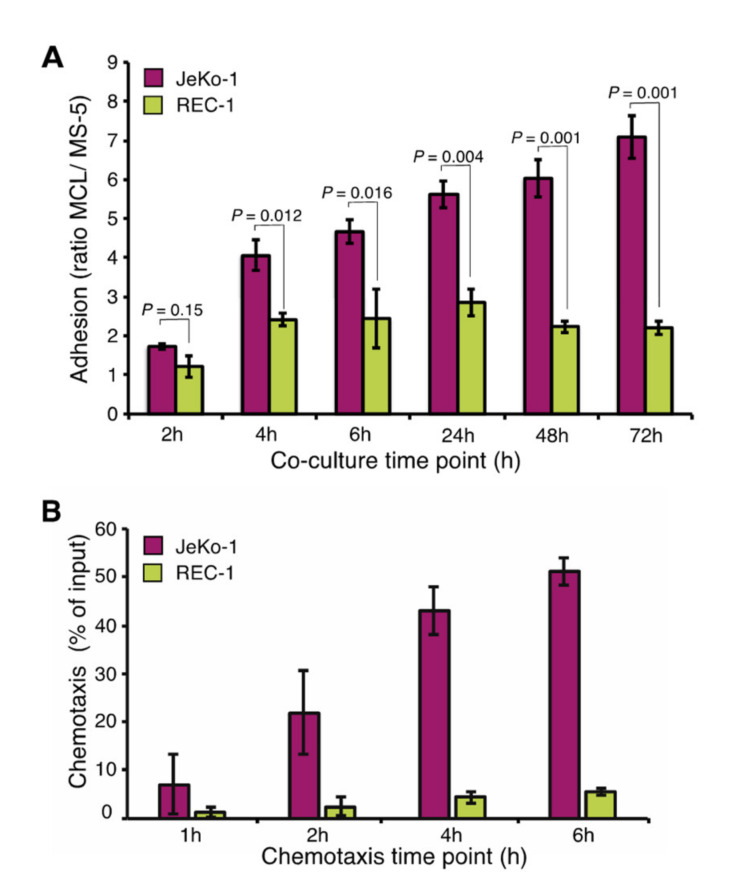
Different adhesion of JeKo-1 and REC-1 MCL cells to stroma. (**A**) Two human mantle cell lymphoma (MCL) cell lines, JeKo-1 and REC-1, were cocultured with MS-5 stromal cells at a 10:1 ratio for different time-periods. The number of adhered MCL cells was quantified after washing the wells as the ratio of MCL cells to stromal cells, using flow cytometry (*n* = means from ≥3 culture wells in three independent experiments). Error bars represent the standard error of the mean (SEM). *p*-values represent statistical significances between two cell lines at different time points. (**B**) Chemotaxis of JeKo-1 (pink) and REC-1 (yellow) cells was measured by a transwell migration assay measuring migration towards stroma-conditioned medium for mentioned time points (*n* = 4 independent experiments). An average of the percentage of input cells is shown, error bars represent SEM. *P* values represent statistical significances between migration towards medium or conditioned medium for each cell line. JeKo-1 *p* values: 0.36, 0.09, 0.004, 0.008 and REC-1 *p* values: 0.38, 0.31, 0.02, 0.007.

**Figure 2 cancers-12-01143-f002:**
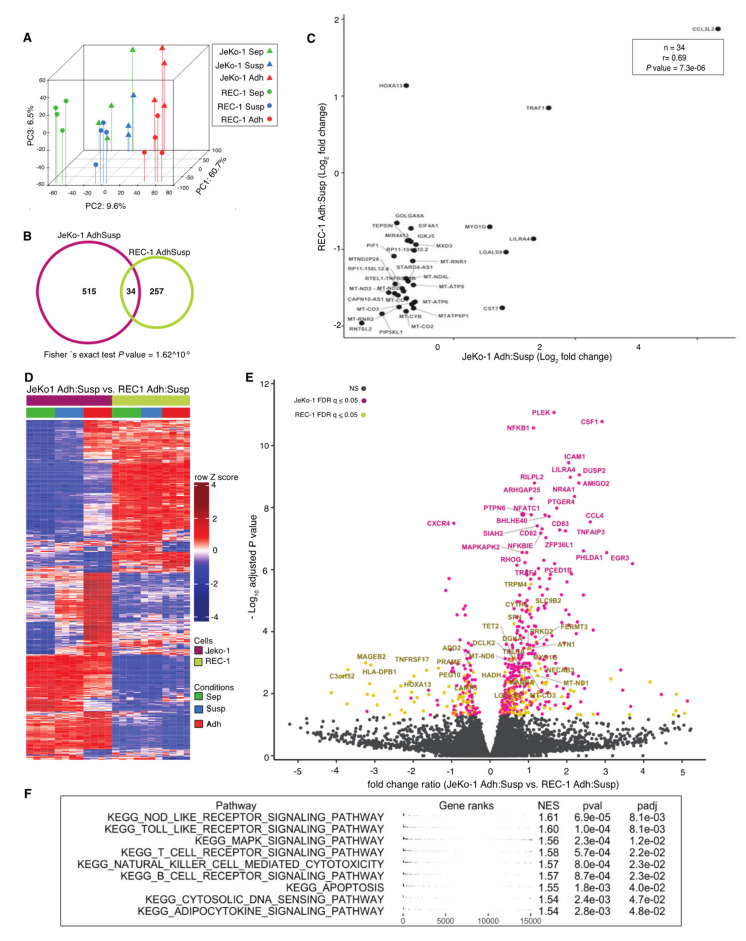
Adhesion to stroma affects global gene expression differently in JeKo-1 and REC-1 cells (**A**) Principle component analysis of genome-wide RNA transcription data from REC-1 (circles) and JeKo-1 (triangles) cells for three different fractions: monocultured cells (Sep: in green), suspension cells within coculture (Susp: blue), and adherent cells within the coculture (Adh: red). (**B**) Venn diagram showing the number of differentially expressed genes between adherent JeKo-1 cells relative to suspension cells (pink circle, false discovery rate (FDR) *q*-value ≤ 0.05, absolute fold change (FC) ≥ 1.5) and for adherent REC-1 cells relative to suspension cells in the coculture (yellow circle, FDR *q*-value ≤ 0.05, FC ≥ 1.5). Thirty-four genes were differentially expressed in both adherent JeKo-1 and adherent REC-1 cells when compared to their relative suspension counterpart. (**C**) Scatter plot of log_2_ transformed FC for the 34 intersect genes from (**B**). (**D**) Heatmap of 590 genes that are differentially expressed in adherent REC-1 cells relative to suspension cells in the coculture compared to adherent JeKo-1 cells relative to suspension (*n* = 4 independent experiments, FDR *q*-value ≤ 0.05). (**E**) Volcano plot representation for the comparison of genes that are differentially expressed between Adh versus Susp JeKo-1 cells and Adh versus Susp REC-1 cells. Pink represents genes with significantly changed transcript levels where the differential of FC can be ascribed to a higher absolute FC in JeKo-1 and yellow represents genes where the absolute FC is higher in the REC-1 cells (FDR *q*-value > 0.05). Grey: genes with nonsignificant transcript level changes. (**F**) Gene set enrichment analysis for gene sets from the KEGG pathway database using the differently regulated gene set (*n* = 590). A positive normalized enrichment score (NES) represents gene sets that were enriched for due to a higher regulation in the JeKo-1 cells and a negative NES for gene sets containing genes with a higher regulation in REC-1. Significantly enriched KEGG pathways are shown and a full table with enriched pathways is available as Appendix A.

**Figure 3 cancers-12-01143-f003:**
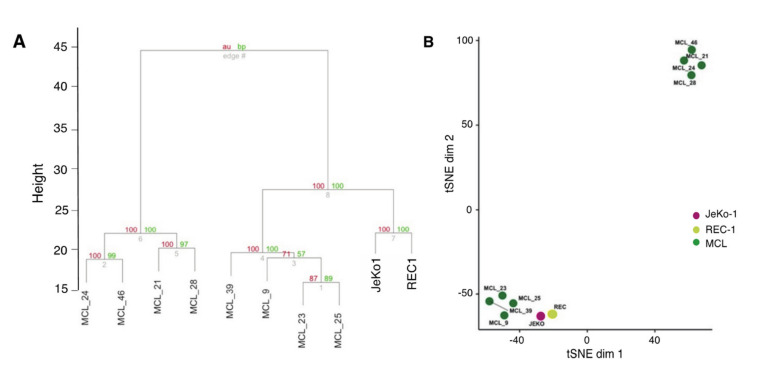
Adhesion-regulated gene expression in cell lines mimics microenvironment-regulated gene expression in MCL patients. (**A**) Dendrogram representation of hierarchical clustering based on log2 fold changes for differentially expressed genes between adherent and suspension JeKo-1 and REC-1 cells and cells of lymph node (LN) versus peripheral blood (PB) origin from MCL (total 684 genes). Clustering only performed using adhesion-dependent differentially regulated genes. (**B**) tSNE representation of data presented in A with perplexity set to 3.

**Figure 4 cancers-12-01143-f004:**
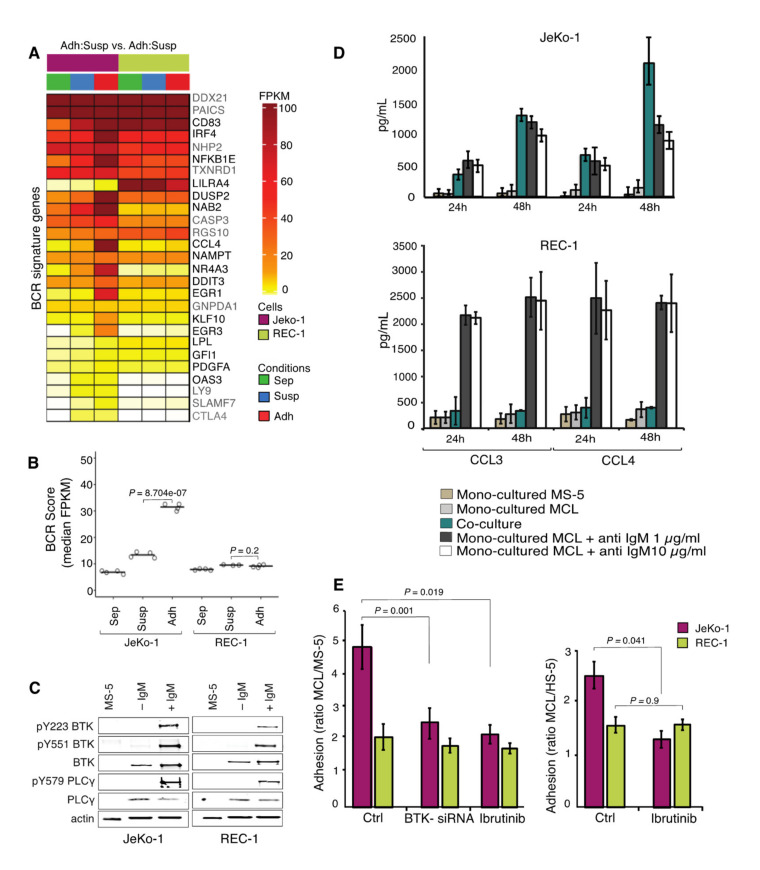
Differential expression differences in B-cell receptor (BCR) signature genes in JeKo-1 and REC-1 cells upon adhesion to stroma. (**A**) Transcript levels for BCR signature genes for three different fractions of JeKo-1 and REC-1 cells, monocultured cells (Sep, in green), suspension cells in coculture (Susp, blue), and adherent cells in coculture (Adh, red). Genes with a significant differential expression are marked in black (FDR *q*-value ≤ 0.05 and absolute fold change ≥ 1.5) (*n* = 4 independent experiments). (**B**) Median FPKM values (Fragments Per Kilobase per Million of reads) for BCR signature genes for mono- and cocultured (Susp and Adh) JeKo-1 and REC-1 cells. *p*-values are calculated using Fisher’s exact test. (**C**) Functional characterization of BTK in JeKo-1 and REC-1 cells using western blot analysis. Cells were activated with anti-human IgM (10 μg/mL) and H2O2 (4 mM) for 5 min at room temperature and whole cell lysate were extracted and subjected to immunostaining with anti-BTK, anti-phosphorylated BTK (Y223 and Y551), anti-PLCγ2, and anti-phosphorylated PLCγ2 (Y579) antibodies. (**D**) CCL3 and CCL4 cytokine levels in conditioned media from mono-cultured and cocultured JeKo-1 and REC-1 cells compared to monocultured JeKo-1 and REC-1 cells in the presence of anti-human IgM following 24 h and 48 h of coculture or anti-IgM stimulation. CCL3 and CCL4 concentrations were measured by sandwich Elisa. Brown: MS-5, Light gray (JeKo-1 or REC-1) cells without anti-human IgM, Turquoise: (JeKo-1 or REC-1) cocultured with MS-5, Dark gray and white: (JeKo-1 or REC-1) with anti-human IgM treatment (1 and 10 μg/mL). (**E**) The ratio of adhered JeKo-1 or REC-1 cells to mouse or human stromal cells was quantified after 4 h of coculture using flow cytometry. Untreated cells, BTK knockdown (BTK-siRNA) and Ibrutinib (0.5 µM) treated cells. (*n* = 3 independent experiments). Error bars represent the SEM Student’s *t*-test was performed and the *p*-values indicate the significance of differences between the number of stromal cell adhered JeKo-1 or REC-1 cells in Ctrl and BTK inhibited cells.

**Figure 5 cancers-12-01143-f005:**
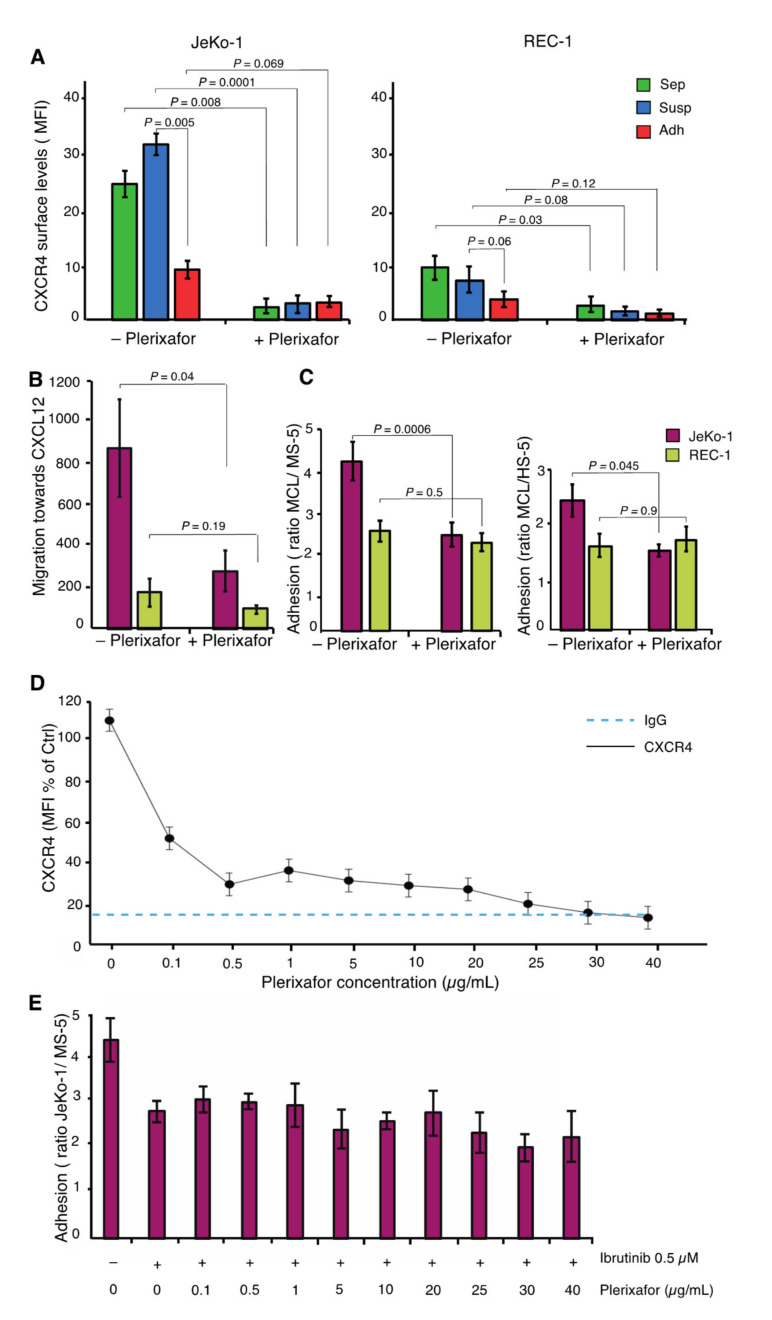
JeKo-1 and REC-1 cells express different cell surface levels of CXCR4 (**A**) Membrane expression level of CXCR4 in three different fractions of JeKo-1 and REC-1 cells without and with Plerixafor treatment (final concentration of 25 µg/mL), presented as mean fluorescent intensity (*n* = 3). (**B**) Chemotaxis of JeKo-1 (pink) and REC-1 (yellow) cells was measured by transwell migration assay towards CXCL12 without and with Plerixafor for 4 h (final concentration of 25 µg/mL). Bars represent the number of migrating cells toward CXCL12 normalized to medium without CXCL12. (**C**) MCL cells were cocultured with either MS-5 or HS-5 stromal cells without and with 25 µg/mL Plerixafor for 4 h and the number of adherent cells was quantified using flow cytometry (*n* = 3 independent experiments and error bars represent SEM). Student’s *t*-test was performed, and the *p* value indicates differences between the number of stromal cells adhered JeKo-1 cells in treated and untreated cells. (**D**) Plerixafor dose–response curve in JeKo-1 cells presented as mean fluorescent intensity (*n* = 3). (**E**) JeKo-1 cells were treated with 0.5 µM Ibrutinib for 1 h, then treated cells were cocultured with MS-5 cells in the absence or presence of different concentrations of Plerixafor for 4 h and the number of adherent cells were quantified using flow cytometry (*n* = 3 independent experiments and error bars represent SEM).

**Figure 6 cancers-12-01143-f006:**
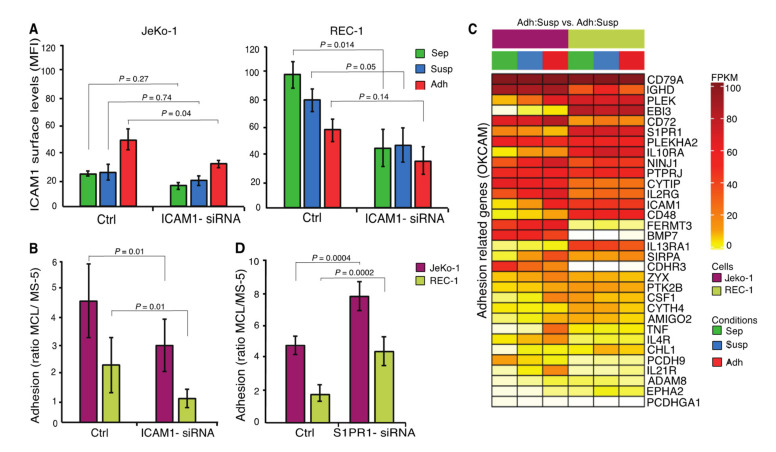
ICAM1 and S1PR1 are important for the adhesion of MCL cells to stroma (**A**) Membrane expression of ICAM1 in three different fractions of JeKo-1 and REC-1 cells without and with ICAM1-siRNA knockdown presented as mean fluorescent intensity. (**B**) MCL cells were cocultured with MS-5 stromal cells for 4 h without and with ICAM1 siRNA knockdown and the number of adherent cells was quantified using flow cytometry. (*n* = 3 independent experiments and error bars represent SEM). (**C**) Transcript levels for significantly differentially expressed genes (FDR *q*-value ≤ 0.05) involved in cell adhesion (Genes retrieved from the cell-adhesion database OKCAM (http://okcam.cbi.pku.edu.cn)) for three different fractions of JeKo-1 and REC-1 cells, monocultured cells (Sep) in green, suspension cells in coculture (Susp) in blue and adherent cells in coculture (Adh) in red. (**D**) MCL cells were cocultured with MS-5 stromal cells for 4 h without and with S1PR1 siRNA knockdown and the number of adherent cells was quantified using flow cytometry. (*n* = 3 independent experiments and error bars represent SEM). Student’s *t*-test was performed and the *p* value indicates differences between the number of stromal cells adhered MCL cells in nontransfected and siRNA-transfected cells.

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
