# Peer review of "Differential B-Cell Receptor Signaling Requirement for Adhesion of Mantle Cell Lymphoma Cells to Stromal Cells"

_cancers, 2020, doi:10.3390/cancers12051143_

Round 1

Reviewer 1 Report

This is an interesting study and would provide more evidence on relevance of cell lines for researchers studying MCL microenvironment.

Few comments

  1. How do you explain lack of CD49d on your GEP data?
  2. Were the MCL patients untreated or treated with BTK inhibitor before?
  3. If the Rec-1 is originating from the LN then they should have higher BCR pathway over-expression compared to PB derived Jecko cell lines, but this was not clear. This difference might explain the differential signaling.
  4. What is the translational relevance of this work, this should be explained.
  5. Why only CCL3 and CCL4 were included, did you investigate IL-17, osteopontin or CSF-1 levels in co-cultures?
  6. Will it make a difference if you have stromal derived macrohages vs stromal derived follicular dendritic T cells?
  7. Do the authors think of adding pexidartinib and evalaute the MFI of CSF-1R, as reported by Papin et al Leukeimai 2018?

Author Response

We would like to thank the reviewers for their suggestions and comments. We have made changes in response, which we believe have improved the study. Our point-by-point responses are detailed in blue below.

Reviewer 1

  1. How do you explain lack of CD49d on your GEP data?

ITGA4 (integrin a4 or CD49D) and ITGB1 (integrin b1 or CD29) are expressed in both JeKo-1 and REC-1 cells (see table below – extracted from Table S1, number represent FPKM). a4b1, for example, plays a role in the migration of MCL cells beneath stromal cells (Kurtova et al. blood (2009) and could together with other adhesion molecules, be important for stromal cell adhesion of JeKo-1 or REC-1 cells in our co-culture. However, integrins do not seem to be regulated during adhesion in our system and in line with the design of the study, we have focused on genes that are regulated. We have modified the text (lines 387-388) to clarify this point.

ensemble

name

length

JeKo-1 mono

JeKo-1 susp

JeKo-1

Adh

REC-1 mono

REC-1 susp

REC-1

Adh

ENSG00000115232

ITGA4

7901

27.051007

24.2401442

21.2217441

14.7210488

12.7097475

12.8071884

ENSG00000150093

ITGB1

6011

27.1571862

27.2580005

28.4741128

22.1099081

21.5510577

19.887967

  1. Were the MCL patients untreated or treated with BTK inhibitor before?

We have checked the publication from which this data was obtained and found that the lymph node biopsies and peripheral blood samples were obtained from patients with previously untreated MCL. This information was added to the revised manuscript (line 198).

  1. If the Rec-1 is originating from the LN then they should have higher BCR pathway over-expression compared to PB derived JeKo1 cell lines, but this was not clear. This difference might explain the differential signaling.

CCL3 and CCL4 protein levels after IgM stimulation were 2-fold higher in mono-cultured REC-1 cells compared to JeKo-1 cells, suggesting a higher potential for BCR signaling in REC-1 cells. Consistently, expression levels of the B-cell receptor subunit CD79A gene are about 2-fold higher in REC-1 compared to JeKo-1 although CD79B gene expression levels were similar in both cell lines (Table S1). In spite of this higher BCR potential, there was no significant induction of BCR signature genes (Fig 4A, B) or CCl3/ CCL4 protein levels (Fig 4D) in Rec-1 cells upon adhesion to stromal cells. This was in strong contrast to the behavior of JeKo-1 cells. We have modified the text to better explain the difference between the potential BCR signaling capacity and the extent to which BCR is engaged upon stromal cell adhesion in the two cell lines (lines 254-255).

  1. What is the translational relevance of this work, this should be explained.

In the revised manuscript, we have reworded the discussion section to highlight the translational relevance of this work (lines 352-357).

  1. Why only CCL3 and CCL4 were included, did you investigate IL-17, osteopontin or CSF-1levels in co-cultures?

CCL3 and CCL4 protein levels were measured in the co-culture primarily to check activation of downstream BCR response at the protein level. However, the RNA-seq data shows that adherent JeKo-1 cells but not REC-1 cells express elevated levels of CSF1 compared to mono-cultured cells (Fig 2E). The levels of osteopontin (SPP1) were not greatly changed in response to adherence and were therefore of less interest for this study (see extract from Table S1 below, numbers represent FPKM). We did not identify any RNAseq reads for IL-17A.

ensemble

name

length

JeKo-1 mono

JeKo-1 susp

JeKo-1

Adh

REC-1 mono

REC-1 susp

REC-1

Adh

ENSG00000118785

SPP1

2321

25.0067577

20.870472

22.9603426

27.7284861

31.189884

31.7719902

ENSG00000184371

CSF1

5418

0.72449584

1.23824031

9.46307797

6.84968904

4.56901805

4.60182301

  1. Will it make a difference if you have stromal derived macrophages vs stromal derived follicular dendritic T cells?

Different cell subtypes participate in the MCL microenvironment including follicular dendritic cells (FDC), mesenchymal stem/stromal cells (MSC) and immune cells e.g. macrophages and T-cells, establishing dynamic interactions with MCL cells to promoting their proliferation and drug resistance (Papin et al. leukemia (2019); T. Lwin, J. Lin et al. Blood (2010); J. Guan et al. Mol Cancer Ther (2018)). In a complete microenvironment the output would be the result of interactions between all relevant cell types. The strength but also limitation of the present study design is that we focus only on interactions between lymphoma cells and stromal cells. In a possible extension of the work it would be possible to introduce other cell types to see how (i) existing effects are affected and (ii) which new effects are created when new cell types are introduced. Thus the difference would be that the system would be more complex but that it would be more difficult to identify effects caused by interactions between specific pairs of cell types. There are a number of design problems to overcome before such systems could be reconstituted. We have added text at the end of the discussion to actualize this possibility (lines 436-437).

  1. Do the authors think of adding pexidartinib and evalaute the MFI of CSF-1R, as reported by Papin et al Leukeimai 2018?

We thank the reviewer for bringing the work of Papin et al. leukemia (2019) to our attention. We have added text in the Discussion to discuss the results of the paper and their possible significance in relation to our own results (lines 404-409).

Reviewer 2 Report

This is an interesting and important study with great potential. The manuscript is extremely well-written and concise. The reported data are of utmost importance. This work definitely will have a noticeable impact. In my view, this work should be further strengthened by conduction some additional bioinformatics analyses.
1) The authors should conduct at least minimal functional analysis of the genes that are differently regulated upon adhesion of the cell lines to stromal cells, as well as genes that are similarly regulated in both cell lines. For example, gene ontology (GO) enrichment analysis should be performed on these gene sets.
2) The authors should also conduct some analysis of intrinsic disorder predisposition of proteins encoded by genes that are differently regulated upon adhesion of the cell lines to stromal cells and genes that are similarly regulated in both cell lines.
3) Some correlations should be analyzed between the levels of predicted intrinsic disorder in these sets of proteins and levels of overexpression of their genes. Similarly, the authors may analyze correlations between disorder predispositions and GO annotated functions.

Author Response

We would like to thank the reviewers for their suggestions and comments. We have made changes in response, which we believe have improved the study. Our point-by-point responses are detailed in blue below.

Reviewer 2

  1. The authors should conduct at least minimal functional analysis of the genes that are differently regulated upon adhesion of the cell lines to stromal cells, as well as genes that are similarly regulated in both cell lines. For example, gene ontology (GO) enrichment analysis should be performed on these gene sets.

GSEA analysis using differently regulated genes in JeKo-1 and REC-1 cells was performed (see Figure 2F and Table S2 but we agree that we should have performed a similar analysis for the set of 34 similarly regulated genes. This has now been done and the results are shown in an additional supplementary table (new Table S3). In relation to this but also elsewhere we have changed the text to better explain the functional analysis that was performed (see lines 143-146,150-153 and 168-170)

  1. The authors should also conduct some analysis of intrinsic disorder predisposition of proteins encoded by genes that are differently regulated upon adhesion of the cell lines to stromal cells and genes that are similarly regulated in both cell lines. 

In revised manuscript, characterization of IDR regions in proteins encoded by genes similarly or differently regulated upon adhesion to stromal cells was performed and showed that the IDR content of similarly regulated genes is significantly lower than for differently regulated genes (Figure S3A), see new text lines 172-182).

  1. Some correlations should be analyzed between the levels of predicted intrinsic disorder in these sets of proteins and levels of overexpression of their genes. Similarly, the authors may analyze correlations between disorder predispositions and GO annotated functions.

In revised manuscript, correlation between IDRs and gene expression levels of adhesion regulated genes that encode them has been performed in both JeKo-1 and REC-1 cells (Figure S3B-3E), (see new text lines 182-191). There was an indication that higher expressed similarly regulated genes tend to have fewer IDRs but this was not established for the larger set of differently regulated genes.

Round 2

Reviewer 1 Report

Thank the authors for adequately answering the questions.

Reviewer 2 Report

All comments were adequately addressed and the manuscript was revised accordingly. I do not have additional recommendations.